# Spatio-temporal evolution and convergence patterns of E-commerce development in China: Regional disparities and policy implications

Yuhuan Wu☯, Haisong Wang☯*, Daqun Guo, Yan Li, Hang Zhang, Tianyuan Shan

Department of Economics and Trade, Hebei University of Water Resources and Electric Engineering, Cangzhou, Hebei, China

☯ These authors contributed equally to this work.

* micromoon@hbwe.edu.cn

## Abstract

The rapid development of e-commerce has become a key driver of economic transformation and regional development in China. However, significant spatial and temporal disparities persist across regions. Understanding these disparities is essential for promoting balanced growth and maximizing the potential of e-commerce in fostering regional economic integration. However, existing studies often focus on isolated temporal trends or spatial patterns, lacking a comprehensive analysis that integrates spatial-temporal dynamics and examines the convergence trends of e-commerce development. This study aims to address these gaps by investigating the spatial-temporal distribution characteristics of e-commerce in China and assessing its convergence trends across different regions. This study utilizes panel data from 31 Chinese provinces spanning the period 2013-2023, employing spatial statistical methods, $\sigma$-convergence and $\beta$-convergence models, and regression analysis to identify the key factors influencing the observed disparities. The study finds that China's e-commerce development is characterized by significant regional differences, the coexistence of polarization and low density, and different regional convergence trends in different regions, which provides an important theoretical basis for formulating differentiated development strategies, strengthening cross-regional infrastructure synergies, and digital technology innovation.

## Introduction

E-commerce, as a pivotal driving force in the digital economy, has profoundly reshaped traditional business models and significantly influenced the distribution of social resources and economic patterns. China, as a global leader in e-commerce, has consistently held the top position in the global ranking of its e-commerce market size, and its contribution to the national economy has been steadily increasing. In April 2024, the Ministry of Commerce (MOFCOM) issued the Three-Year Action Plan for Digital Commerce (2024-2026), which proposes to "better promote the digital transformation of various fields of commerce,

**Data availability statement:** All relevant data are within the manuscript and its Supporting information files.

**Funding:** This research was funded by the 2025 Hebei Province Higher Education Teaching

Reform Research and Practice Project (Project Name: Research on the Innovation of the "Learning-Practice-Competition-Reflection-Application-Innovation" Talent Cultivation Model for Cross-Border E-Commerce Undergraduate Programs)(2025GJJG396, received by A.P. Haisong Wang); This research was funded by Science Research Project of Hebei Education Department (BJS2024097, received by Dr. Yuhuan Wu); This research was Supported by the Fundamental Research Funds for the Hebei University of Water Resources and Electric Engineering (SYKY2502, Dr. Yuhuan Wu); This research was funded by the Ministry of Education, Industry-University Cooperation Collaborative Education Project (230825052507181, received by Dr. Yuhuan Wu). The funders had no role in study design, data collection and analysis, decision to publish, or preparation of the manuscript.

**Competing interests:** The authors have declared that no competing interests exist.

empower economic and social development, and serve to build a new development pattern." However, as China's e-commerce sector experiences rapid growth, the disparities in its development across different regions are becoming increasingly evident, underscoring the challenges in achieving coordinated regional advancement in the digital economic era. From a temporal perspective, China's e-commerce evolution can be characterized by three distinct phases: an initial period of rapid growth, a subsequent phase of structural optimization, and the present phase of high-quality development [1]. The spatial distribution of e-commerce, growth rate, and resource utilization efficiency have undergone significant dynamic changes at each stage. Guided by national policies and with the gradual popularization of digital infrastructure, the e-commerce potential of the central and western regions has been realized, with the development speed of some regions exceeding that of the eastern coastal regions [2]. This dynamic evolution of regional development differences has led to the complexity of the spatial and temporal distribution characteristics of e-commerce in China and the evolution of the law. Consequently, examining the characteristics of the spatial and temporal distribution of e-commerce in China, as well as its evolutionary patterns, has emerged as a pivotal subject in academic research and policy formulation.

Furthermore, an examination of the convergence of development levels among regions can elucidate whether the development of e-commerce exhibits a propensity towards equilibrium or not, in addition to the underlying driving mechanisms that facilitate this process. This provides a theoretical foundation for the realization of coordinated regional economic development. Consequently, an exhaustive examination of the spatial and temporal distribution characteristics of e-commerce in China and its convergence is of paramount theoretical significance. Moreover, it can furnish practical references for the optimization of policy design and the promotion of balanced regional development.

A substantial body of exploratory research has been conducted on the spatial distribution of e-commerce and its influencing factors by scholars both domestically and internationally. This research offers valuable insights into the spatio-temporal dynamic characteristics of e-commerce development. In the context of international studies, scholars largely concur that the spatial distribution of e-commerce is significantly correlated with the level of economic development. They also identify a variety of factors that influence this distribution. For instance, Geng, J et al. (2020) have noted that e-commerce is predominantly concentrated in economically developed regions, and its agglomeration effect is strongly associated with the regional economic level and Internet penetration rate [3]. Bădîrcea RM et al. (2022) have conducted a study on the spatial distribution pattern of e-commerce in EU countries, highlighting that infrastructure development (e.g., logistics network, broadband coverage) is a pivotal Constraints of infrastructure and policy environment in developing countries [4]. These studies offer valuable insights into the characteristics of the global spatial distribution of e-commerce. However, the majority of these studies focus on developed economies or inter-country comparisons, and there is a need for more analyses on the dynamic evolution of e-commerce and the convergence of China's inter-regional e-commerce.

Domestically, scholars have conducted in-depth studies on the regional distribution characteristics of e-commerce in China and the factors that influence it. Wei (2019) used the center of gravity analysis to explore the spatial migration path of e-commerce development in China, revealing that the center of gravity of e-commerce has gradually shifted from the eastern coastal region to the central and western inland regions [5–7]. However, the eastern region remains the core. Wang et al. (2023) employed spatial autocorrelation analysis to elucidate the spatial agglomeration characteristics of e-commerce in China and its hotspot regional distribution law. The study revealed that regional economic level, policy support, and logistics infrastructure construction are pivotal factors influencing the spatial distribution of

e-commerce [8–10]. Building upon these findings, Tang et al. (2020) identified the driving factors of the spatial distribution of e-commerce in China. They employed the geodetector model to conclude that variables such as industrial base, city size, and consumer market capacity play an important role in the development of e-commerce [11–15].

Despite the valuable insights these studies provide into the regional distribution characteristics of e-commerce, and while some recent works have begun to address spatio-temporal dynamics or convergence in other economic sectors [16–18], they predominantly emphasize static analysis and often overlook the crucial dynamic changes in the development of e-commerce and its inherent convergence characteristics. Moreover, extant studies are typically constrained to the role analysis of single factors, or apply non-spatial regression models [19,20], thereby overlooking the complex interplay of multiple determinants, the critical role of regional heterogeneity, and intricate influence mechanisms within e-commerce development. Notably, a comprehensive framework integrating dynamic spatio-temporal evolution, convergence analysis, and the complex spatial interactions of multiple influencing factors remains largely unexplored, particularly within the context of large developing economies like China. Addressing these significant limitations, this study proposes a more comprehensive and innovative analytical framework, distinctly differentiating itself from prior research in several key aspects. Firstly, unlike studies primarily offering cross-sectional snapshots or limited temporal analyses of e-commerce, this research commences with a rigorous analysis of the dynamic changes in spatio-temporal distribution characteristics. Employing various spatio-temporal statistical methods, such as spatial autocorrelation analysis, it comprehensively portrays the spatio-temporal evolution law of China's e-commerce development, and explores its spatial expansion trend and regional agglomeration characteristics. This temporal depth and spatial granularity significantly advance our understanding beyond static representations. Secondly, while some economic convergence studies exist for other industries or macro-economic indicators, there is a notable paucity of research systematically assessing the convergence of e-commerce development levels, particularly within a developing economy context where regional disparities are often pronounced and dynamic. This study explicitly employs $\sigma$-convergence and $\beta$-convergence models to bridge this critical gap, systematically assessing the convergence of e-commerce development levels among regions in China. It focuses on analyzing the characteristics of the convergence of development levels among different regions and their driving mechanisms [21–25]. Finally, and critically for enhancing originality, diverging from studies that analyze factors in isolation or employ non-spatial regression models, this study explores the influencing factors of spatial and temporal distribution and convergence of e-commerce with the Spatial Durbin Model (SDM). This sophisticated spatial econometric approach allows for an unparalleled exploration of complex, interactive, and heterogeneous effects of policy support, economic level, digital infrastructure, and other factors, including crucial spatial spillover effects often overlooked in non-spatial analyses. Specifically, it moves beyond merely identifying direct effects to analyzing how these factors' impacts vary across different regions and how neighboring regions influence each other, providing a nuanced understanding of regional heterogeneity and complex influence mechanisms in e-commerce development, which is particularly relevant for the policy implications in a vast and diverse country like China.

The following challenges were encountered during the course of this study: Firstly, difficulties arose during the acquisition and processing of the data. E-commerce related data is comprised of multi-dimensional indicators, including transaction volume, Internet penetration rate, and logistics facilities coverage. It should be noted that these data sources differ in terms of statistical caliber and time span. Secondly, the precise portrayal of spatio-temporal dynamic features necessitates the integration of multiple statistical and econometric analysis

methods. Finally, the analysis of the driving mechanism of regional differences necessitates the consideration of spatial dependence and heterogeneity among variables.

In order to address the aforementioned research challenges and attain the research objectives, this study proposes the following research approach and methodologies (as illustrated in Fig 1): First, e-commerce data from 31 provincial-level administrative regions in China from 2000 to 2023 will be collected and organized. This data will include e-commerce turnover, Internet penetration rate, coverage of logistic facilities, and the intensity of policy support, among other metrics. Using these data, a systematic evaluation index system of e-commerce development level will be constructed. Secondly, we will explore the spatial correlation of e-commerce development in China and the distribution pattern of hotspot areas based on global and local Moran's I index. We will then test the convergence trend of e-commerce development level among regions using $\beta$-convergence and $\sigma$-convergence models. Finally, we will further analyze the convergence patterns and difference characteristics among East, Central, West and major urban agglomerations. Finally, we employ the spatial Durbin model to quantitatively assess the comprehensive impact of policy support, economic level, infrastructure construction, and other factors on the spatial and temporal distribution of e-commerce and its convergence. We also analyze the heterogeneous role of inter-regional driving factors in depth.

The research approach and methodology outlined above serve to enhance the existing theoretical framework concerning e-commerce regional development discrepancies and the laws governing dynamic evolution. This enhancement provides valuable practical references for the optimization of China's e-commerce development policies, thereby facilitating

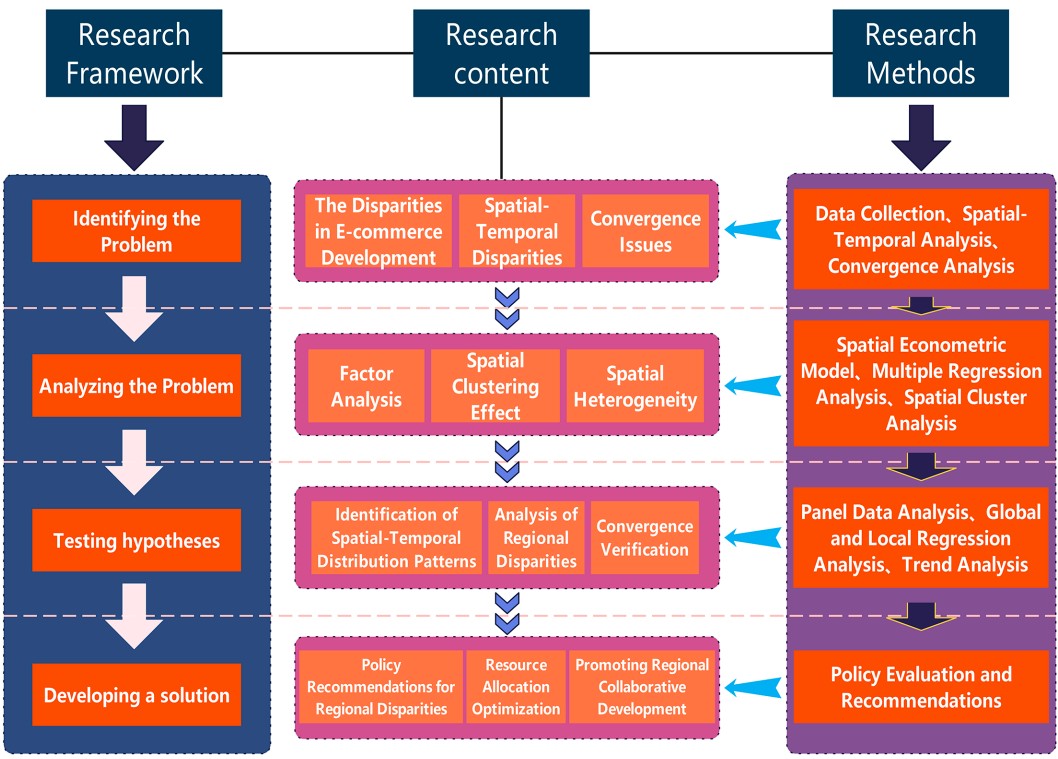

**Fig 1. Research approach and methodology.**

the enhancement of regional development through a balanced approach. Concurrently, the study's findings offer significant policy insights, particularly with regard to the promotion of China's e-commerce in the central and western regions, and the narrowing of the existing regional development gap.

# Data and methodology

## Data sources and preprocessing

Data pertaining to the development of e-commerce is derived from authoritative statistics, including the China Statistical Yearbook, the China Internet Development Report, and the China E-commerce Report. These statistics encompass pivotal indicators such as e-commerce turnover, Internet penetration rate, the number of mobile payment users, and the progression of cross-border e-commerce.

In order to objectively evaluate the comprehensive development level of e-commerce, this study introduces a series of economic and social development indicators, as shown in Table 1. The data come from the National Bureau of Statistics, provincial statistical yearbooks, and the China Regional Economic Development Report. Specifically, the infrastructure category encompasses the telephone penetration rate, the number of domain names, the number of web pages, and the number of Internet broadband access ports. The transaction scale category includes e-commerce sales and e-commerce purchases. The enterprise development category incorporates the proportion of enterprises engaging in e-commerce transactions and the number of websites owned by the enterprises; the talent and innovation category includes the wage level of e-commerce employees and the proportion of e-commerce employees; and the logistics distribution category includes the national share of express delivery business volume. The category of logistics and distribution includes the proportion of express delivery business volume in the country.

In order to analyze the spatial distribution characteristics of e-commerce in China and its dynamic evolution, this study also cites geographic information data. This part of the data comes from the National Geographic Information Public Service Platform (Map World). This platform contains the boundary vector data of provincial administrative regions, the spatial coordinates of major cities, and the distribution of transportation networks. These data reveal the spatial agglomeration characteristics of e-commerce development and its evolution pattern.

In the context of data processing, this study implemented a multifaceted approach to ensure the comparability and integrity of the data. Initially, indicators exhibiting disparities in statistical metrics across different periods were harmonized and standardized to mitigate the influence of statistical inaccuracies on the research outcomes. Subsequently, this study employed time-series linear interpolation to impute missing values. The rationale for selecting this method was primarily based on the understanding that the research indicators typically exhibit continuity and trending patterns in the temporal dimension, and linear interpolation is capable of effectively capturing linear patterns of change between adjacent time points. It is crucial to acknowledge that the interpolation process inherently introduces a degree of uncertainty. The main potential bias of time-series linear interpolation lies in its assumption of linear progression between data points, which may smooth out or overlook actual non-linear fluctuations during the missing periods, thereby potentially underestimating the true data variability. Despite these limitations, this imputation process was deemed crucial for ensuring the completeness and integrity of the dataset, as well as the feasibility of subsequent analyses. Ultimately, these preprocessing steps ensured the representativeness and scientific robustness of the final dataset. Finally, to ensure the validity of the spatio-temporal analysis,

**Table 1. Indicator system of China's comprehensive e-commerce development level.**

| Tier1 indicators | Tier 2 indicators | Nature of the Indicator | Data sourcesd |
|---|---|---|---|
| Infrastructure Category | Telephone penetration rate (units/100 people)X1 | Positive | China Statistical Yearbook |
| | Domain name count (in 10,000s)X2 | Positive | China Statistical Yearbook |
| | Web page count (in 10,000s)X3 | Positive | China Statistical Yearbook |
| | Broadband Internet access ports (in 10,000s)X4 | Positive | China Statistical Yearbook |
| Transaction size Category | E-commerce sales (in 100 million Yuan)X5 | Positive | China Statistical Yearbook |
| | E-commerce procurement value (in 100 million Yuan)X6 | Positive | China Statistical Yearbook |
| Enterprise development Category | Proportion of enterprises with e-commerce activities(%)X7 | Positive | China Statistical Yearbook |
| | Number of enterprise-owned websites(number)X8 | Positive | China Statistical Yearbook |
| Talent and Innovation Category | Salary level of e-commerce practitionersX9 | Positive | China Statistical Yearbook |
| | Employed persons in urban units in information transmission, software and information technology services/Employed persons in urban units(%)X10 | Positive | China Statistical Yearbook |
| Logistics and Distribution Category | Percentage of national express delivery volume(%)X11 | Positive | China Statistical Yearbook |

this study performed a spatial matching operation on all the data, combining the e-commerce turnover and related indicators with the geographic boundary data of the corresponding regions.

## Analytical framework

The present study employs a multifaceted analytical framework to investigate the spatio-temporal distribution characteristics and convergence analysis of e-commerce development in China. This analytical framework integrates statistical, spatial, and econometric methods. The framework is composed of three core components, as illustrated in Fig 2. The first component is spatio-temporal pattern analysis. The second component is convergence test. The third component is spatial heterogeneity analysis.

## Methodology

**Entropy method.** The entropy method is an objective empowerment method that is frequently employed in the comprehensive evaluation of multiple indicators. This method determines the weights of indicators by calculating the information entropy of each indicator. The larger the entropy value, the smaller the amount of information of the indicator is, and the smaller the impact on the comprehensive evaluation. Conversely, the smaller the entropy value, the larger the amount of information of the indicator is, and the larger the impact on the comprehensive evaluation [26].

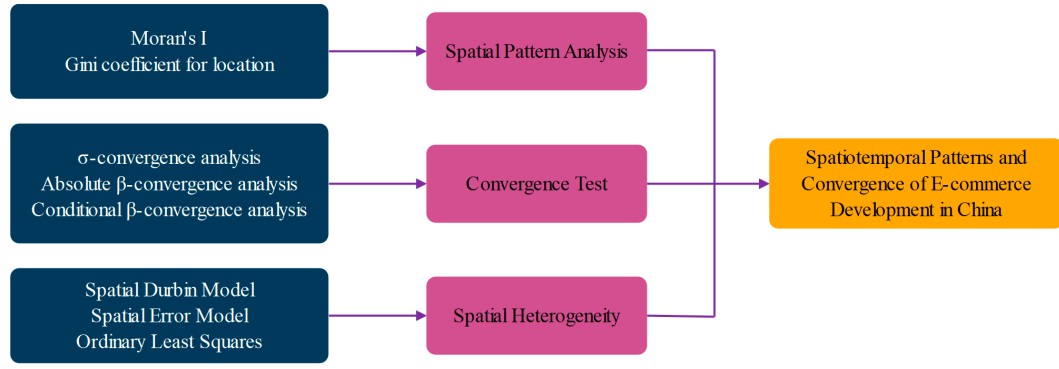

**Fig 2. Analytical framework.**

The fundamental steps of the entropy method are as follows:

1) Constructing the raw data matrix

It is hypothesized that there are m evaluation objects and n evaluation indicators. The raw data moments are therefore as follows:

$$X = \left(x_{ij}\right)_{m \times n} \tag{1}$$

Where: $x_{ij}$ denotes the value of the $i$ object on the $j$ indicator.

2) Data standardization

Positive indicator:

$$p_{ij} = \frac{x_{ij} - \min(x_j)}{\max(x_j) - \min(x_j)} \tag{2}$$

Negative indicator:

$$p'_{ij} = \frac{\max\left(x_j\right) - x_{ij}}{\max\left(x_j\right) - \min\left(x_j\right)} \tag{3}$$

Where:

$x_{ij}$ is the value of indicator $j$;

$\min\left(x_j\right)$ is the minimum value for indicator $j$;

$\max\left(x_j\right)$ is the maximum value for indicator $j$;

$p_{ij}$ and $p'_{ij}$ represent standardized values for positive and negative indicators, respectively.

3) Calculate the entropy value of each indicator

Calculate the weight of the $i$ sample under the $j$ indicator:

$$p_{ij} = \frac{x_{ij}}{\sum_{i=1}^{m} x_{ij}} \tag{4}$$

Calculate the information entropy of this indicator:

$$e_j = -k \sum_{i=1}^{m} p_{ij} \ln p_{ij} \tag{5}$$

Where: $k = \frac{1}{lnm}$ is a normalization factor that ensures that the entropy value is between 0 and 1.

4) Calculate the entropy weights

Calculation of the coefficient of variation for the indicator:

$$d_j = 1 - e_j \tag{6}$$

Where:

$d_j$ is the coefficient of variation for the $j$ indicator.

Calculate the weight of the $j$ indicator:

$$w_j = \frac{d_j}{\sum_{j=1}^{n} d_j} \tag{7}$$

5) Calculate the composite score

$$S_i = \sum_{j=1}^{n} w_j p_{ij} \tag{8}$$

Where:

$S_i$ is the composite score for the $i$ evaluator.

**Dagum Gini coefficient.** The Dagum Gini coefficient is an improvement on the traditional Gini coefficient proposed by Camilo Dagum and is primarily used to measure income or wealth inequality [27]. Compared to the traditional Gini coefficient, the Dagum Gini coefficient is able to provide a more nuanced picture of imbalances in the distribution of income or wealth, especially inequality between groups and inequality within groups.

1) Improvement of the Dagum Gini coefficient

The Dagum Gini coefficient can be expressed as:

$$G = G^w + G^b + G^t \tag{9}$$

Where:

$G^w$ : Within-group inequality;

$G^b$ : Between-group inequality;

$G^t$ : Transvariation inequality.

2) Calculation of the Dagum Gini coefficient It is assumed that a dataset contains $q$ income groups, with $n_k$ individuals in each group, and an income level of $x_{ik}$ :

$$G = \sum_{k=1}^{q} \sum_{h=1}^{q} G_{kh} \tag{10}$$

Where:

$$G_{kh} = \frac{1}{2n_k n_h} \sum_{i=1}^{n_k} \sum_{j=1}^{n_h} \left| x_{ik} - x_{jh} \right| \tag{11}$$

Subsequent to this initial decomposition, Dagum puts forward a series of equations that govern inequality Within-group, inequality Between-group, and inequality Transvariation.

Within-group inequality:

$$G^w = \sum_{k=1}^{q} p_k G_k \tag{12}$$

Where:
$p_k$ is the income share of group $k$;
$G_k$ is the Gini coefficient for the group.
Between-group inequality:

$$G^b = \sum_{k=1}^{q} \sum_{h=1}^{q} p_k p_h \left| \frac{\mu_k - \mu_h}{\mu} \right| \tag{13}$$

Where:
$\mu_k$ denotes the mean income of group $k$;
$\mu_h$ denotes the mean income of group $h$;
$\mu$ denotes the mean income of all samples.
Transvariation inequality:

$$G^t = \sum_{k=1}^{q} \sum_{h=1}^{q} p_k p_h T_{kh} \tag{14}$$

Where:
$T_{kh}$ is the Transvariation Overlap Index.

The Dagum Gini coefficient is a statistical tool that can accurately identify disparities in the development of e-commerce among different geographical regions. By constructing an e-commerce development index system, quantifying the overall development level of e-commerce in each region using the entropy value method, and calculating the Dagum Gini coefficient, we are able to examine and analyze the spatial variations among the four primary regions: East, Central, West, and Northeast. This analysis also encompasses the dynamic evolution of these regions over time [28].

**Moran's I.** Moran's Index (Moran's I) is a pivotal indicator employed in spatial statistics to ascertain the presence of autocorrelation—that is, the existence of a correlation between the values in one region and its neighboring regions. This index enables the assessment of the extent to which a variable exhibits spatial aggregation, in addition to unveiling spatial patterns such as agglomeration and dispersion. In this paper, the Moran index is used to assess the presence of spatial autocorrelation between e-commerce development levels across different regions [29,30]. This analysis aims to unveil inter-regional variability and identify agglomeration trends.

(1) Global Moran's I

The Global Moran's I is a statistical indicator that is employed to measure the spatial autocorrelation of data within the entire study area. It assists in determining whether a phenomenon (e.g., the level of e-commerce development) is distributed in a random manner, is clustered, or is discretely distributed. The Global Moran's I is calculated as follows:

$$I = \frac{n}{S_0} \cdot \frac{\sum_{i=1}^{n} \sum_{j=1}^{n} w_{ij} (x_i - \bar{x})(x_j - \bar{x})}{\sum_{i=1}^{n}(x_i - \bar{x})^2} \tag{15}$$

Where:

$n$ is the number of regions, that is to say, the sample size;

$x_i$ is the observed value for region $i$ (e.g., e-commerce development index);

$x_j$ is the observed value for region $j$;

$\bar{x}$ is the arithmetic mean of all regional observations;

$w_{ij}$ is the element of the spatial weight matrix, which represents the spatial relationship between regions $a$ and $b$. This study employs an adjacency matrix for its analysis.

$S_0$ is defined as the total of all elements contained within the spatial weight matrix, $S_0 = \sum_{i=1}^{n} \sum_{i=1}^{n} w_{ij}$

The presence of $i > 0$ indicates positive spatial autocorrelation, signifying that the e-commerce development levels of neighboring regions exhibit similarity and a clustering effect. Conversely, if $i < 0$ is present, it denotes negative spatial autocorrelation, indicating a substantial difference in the e-commerce development levels of neighboring regions. e-commerce development level of neighboring regions, which manifests itself as a spatial dispersion; and if $i = 0$, it means a spatial random distribution, which indicates that there is no obvious tendency of clustering or dispersing of the e-commerce development level of neighboring regions.

The Global Moran's I can be utilized to assess the agglomeration of different provinces or regions in China in terms of e-commerce development. This approach enables the determination of whether e-commerce has formed a centralized or homogeneous distribution in specific regions or groups of regions.

(2) Local Moran's I

The Local Moran's I is a statistical tool that is primarily employed to discern the local spatial autocorrelation present within each region. This index facilitates the identification of distinct autocorrelation patterns within a specific region, thereby enabling the differentiation of various types of spatial relationships. The formula employed to calculate the localized Moran Index is as follows:

$$I_i = \frac{(x_i - \bar{x})}{S^2} \sum_{j=1}^{n} w_{ij}(x_j - \bar{x}) \tag{16}$$

Where: $S^2 = \frac{\sum_{j=1}^{n}(x_j - \bar{x})^2}{n-1}$ is defined as the variance of all observations.

The local Moran's index, in contrast to the global Moran's index, is employed to discern particular spatial patterns. For instance, the High-High (HH, Hot Spot) configuration signifies regions of high values that are encircled by regions of high values, thereby engendering a clustering effect. Similarly, the Low-Low (LL, Cold Spot) configuration denotes regions of low values that are encircled by regions of low values, resulting in a clustering effect. The third type is the High-Low (HL) configuration, in which high-value regions are surrounded by low-value regions, manifesting as isolated high-value points. The fourth type is the

Low-High (LH) configuration, in which low-value regions are surrounded by high-value regions, manifesting as isolated low-value points.

**Convergence model.** The convergence model is derived from economic growth theory, which is mainly based on the neoclassical growth model and endogenous growth theory [31,32]. The basic idea is that under certain economic environment and policy conditions, the economic development levels of different regions or countries will gradually tend to equalize over time, i.e., the economic gap may show a tendency to narrow, thus achieving a certain degree of convergence. Convergence analysis generally includes $\sigma$-convergence and $\beta$-convergence.

(1) $\sigma$-convergence

$\sigma$-convergence refers to whether the dispersion of an economic variable (e.g. e-commerce development index, etc.) across regions or countries is decreasing over time. The standard deviation (SD) or coefficient of variation (CV) is usually used as a measure. If the standard deviation or coefficient of variation shows a decreasing trend, it indicates that the economic development levels of different regions are converging and there is $\sigma$-convergence.

The following steps are involved in the $\sigma$-convergence analysis:

1) Calculating the mean value

$$\bar{X}_t = \frac{1}{N} \sum_{i=1}^{N} X_{i,t} \tag{17}$$

Where:

$N$ denotes the total number of regions or countries in the sample.

2) Calculating the standard deviation

$$\sigma_t = \sqrt{\frac{1}{N} \sum_{i=1}^{N} \left( X_{i,t} - \bar{X}_t \right)^2} \tag{18}$$

Where:

$X_{i,t}$ is the economic variable (e.g., the e-commerce development index) for the $i$ region or country at time $t$;

$\bar{X}_t$ is the mean value for all regions at time $t$.

3) Calculating the coefficient of variation

$$CV_t = \frac{\sigma_t}{\bar{X}_t} \times 100\% \tag{19}$$

Convergence is indicated by a decrease in standard deviation or coefficient of variation over time. Conversely, an increase or maintenance of standard deviation or coefficient of variation indicates absence of convergence, and the possibility of divergence exists.

(2) $\beta$-convergence

$\beta$-convergence is a statistical measure of economic development that quantifies the rate of growth and convergence of regions with different levels of economic development. In essence, the analysis of $\beta$-convergence examines the dynamic adjustment process of the development level between regions. The presence of $\beta$-convergence in a region is indicated by a negative correlation between its economic growth rate and its initial development level. That is, if a region with a lower initial development level exhibits a faster growth rate, then the presence of $\beta$-convergence can be deduced. The assumption of $\beta$-convergence postulates that, under specific conditions, regions with lower development levels can attain faster growth rates and

eventually attain a stable, long-term equilibrium state. The aforementioned concept of $\beta$-convergence is primarily classified into two distinct categories: absolute $\beta$-convergence and conditional $\beta$-convergence.

1) Absolute $\beta$-convergence

The absolute $\beta$-convergence is predicated on the assumption that all regions possess an identical long-run equilibrium level, with growth rates being contingent solely on initial levels. In the event that a region commences at a lower level but exhibits accelerated growth, it possesses the capacity to ultimately converge with other regions. The regression model is as follows:

$$\frac{1}{T}\ln\left(\frac{X_{i,T}}{X_{i,0}}\right) = \alpha + \beta \ln\left(X_{i,0}\right) + \varepsilon_i \tag{20}$$

Where:

$X_{i,0}$ denotes the economic variables (e.g., e-commerce development index, GDP per capita, etc.) in the initial year(0) for the $i$ region;

$X_{i,T}$ denotes the economic variables of the $i$ region in the termination year(T);

$\frac{1}{T}\ln\left(\frac{X_{i,T}}{X_{i,0}}\right)$ is used to denote the average annual growth rate over a period of years;

$\ln(X_{i,0})$ is defined as the logarithm of the economic variable in the initial year;

$\alpha$ and $\beta$ represent regression coefficients, $\varepsilon_i$ is designated as the error term.

In the event that the parameter $\beta$ is found to be less than zero and is deemed to be statistically significant—that is to say, growth rates are observed to be higher in areas where initial levels are lower—then it can be concluded that there is absolute $\beta$ convergence. Conversely, if $\beta$ is found to be greater than zero or is not deemed to be statistically significant, then it can be concluded that there is no convergence and there may even be a divergence trend.

2) Conditional $\beta$-convergence

The conditional $\beta$-convergence enables disparate regions to possess disparate long-run equilibrium levels (e.g., due to discrepancies in policy, industry structure, infrastructure, etc.). Subsequent to controlling for these factors, if $\beta$ remains significantly negative, it signifies that under analogous development conditions, regions with less advanced economies continue to exhibit faster growth. The regression model is as follows:

$$\frac{1}{T}\ln\left(\frac{X_{i,T}}{X_{i,0}}\right) = \alpha + \beta \ln\left(X_{i,0}\right) + \sum_{k=1}^{n} \gamma_k Z_{i,k} + \varepsilon \tag{21}$$

Where:

$Z_{i,k}$ is a control variable that may affect growth. Consistent with previous studies, this paper selected four control variables: per capita regional GDP(ln_gdp), local fiscal expenditure on transportation(ln_finance), the year-end urban population share by region(ln_county), and the average number of enrolled students at all levels of education per 100,000 population(ln_edu).

If $\beta < 0$ and statistically significant, it suggests that, after controlling for other factors, economic development still exhibits a convergence trend, indicating the presence of conditional $\beta$-convergence. Conversely, if $\beta \geq 0$ or is not statistically significant, it implies that regions with lower development levels have not achieved faster growth.

The degree of e-commerce development exhibits significant variations across different regions (e.g., eastern coastal regions versus central and western regions) due to various factors, including Internet penetration, consumer purchasing power, and logistics systems. The convergence model offers a means to analyze the temporal dynamics of these variations,

determining whether they are converging or maintaining their distinct characteristics. By examining the data on e-commerce sales and the number of users in each province, it is possible to assess the regional disparity of e-commerce on a national scale.

## Results

### Measurement of overall e-commerce development in China

**Descriptive statistics.** As illustrated in Table 2, the development patterns of the 31 provinces, municipalities, and autonomous regions in mainland China have undergone dynamic shifts between 2013 and 2023 in the relevant fields. The disparities between regions have been progressively diminishing, thereby enhancing the overall balance of development. Nevertheless, the characteristics of data distribution in different years exhibit variability. The disparities in development speed and level among regions persist, influenced by a multitude of factors. However, the distribution of data exhibits variability from year to year. A multitude of factors contribute to the heterogeneity in the pace and level of development among different regions. Despite these variations, there has been an observed shift towards a more balanced and coordinated approach in the overall development trajectory.

**Spatial distribution characteristics of China's overall e-commerce development level.** This paper proposes a methodology for assessing the overall e-commerce development level of Chinese provinces (with the exception of Hong Kong, Macao, and Taiwan) from 2013 to 2023. The proposed methodology utilizes an entropy value method to quantify the development level, as illustrated in Fig 3.

As illustrated in (A), the regions that exhibited higher levels of e-commerce development in 2013 were predominantly located in the eastern coastal areas, particularly Shanghai, Jiangsu, Zhejiang, and Guangdong. The darker color of these regions indicates that their e-commerce development level is between 0.456601 and 0.778000. In contrast, the central and western regions demonstrate a comparatively lower level of e-commerce development, as evidenced by the lighter color of their regions, which are concentrated between 0.000000 and 0.047700. This observation suggests that in 2013, there was a discernible regional disparity in the advancement of e-commerce in China, with the eastern region at the forefront and the central and western regions exhibiting a comparatively lagging position.

In 2018 (B), the level of e-commerce development in the eastern coastal regions exhibited a persistent high trend, accompanied by an increase in the depth of color, suggesting a further escalation in the e-commerce development index within these regions. Notably, certain regions have attained an interval of 0.409201 - 0.689800, indicating a significant advancement

**Table 2. Descriptive statistics of entropy method results.**

| Year | Max | Min | Mean | Std.dev. | Median | Kurtosis | Skewness |
|------|--------|--------|--------|----------|--------|----------|----------|
| 2013 | 0.7780 | 0.0302 | 0.1683 | 0.1897 | 0.0830 | 3.3710 | 2.0000 |
| 2014 | 0.7818 | 0.0317 | 0.1815 | 0.1963 | 0.1020 | 3.3260 | 1.9990 |
| 2015 | 0.7750 | 0.0411 | 0.1781 | 0.1874 | 0.1090 | 3.9870 | 2.1060 |
| 2016 | 0.7349 | 0.0385 | 0.1794 | 0.1873 | 0.1120 | 3.3290 | 1.9540 |
| 2017 | 0.6999 | 0.0379 | 0.1706 | 0.1763 | 0.1210 | 3.4490 | 2.0030 |
| 2018 | 0.6898 | 0.0369 | 0.1655 | 0.1716 | 0.1100 | 3.7560 | 2.0290 |
| 2019 | 0.7168 | 0.0356 | 0.1686 | 0.1764 | 0.1130 | 4.1080 | 2.1070 |
| 2020 | 0.7057 | 0.0383 | 0.1719 | 0.1772 | 0.1220 | 3.8480 | 2.0790 |
| 2021 | 0.7350 | 0.0368 | 0.1737 | 0.1801 | 0.1160 | 4.0310 | 2.1090 |
| 2022 | 0.7386 | 0.0318 | 0.1681 | 0.1803 | 0.1100 | 4.0980 | 2.1280 |
| 2023 | 0.6827 | 0.0274 | 0.1618 | 0.1679 | 0.1050 | 3.6600 | 2.0050 |

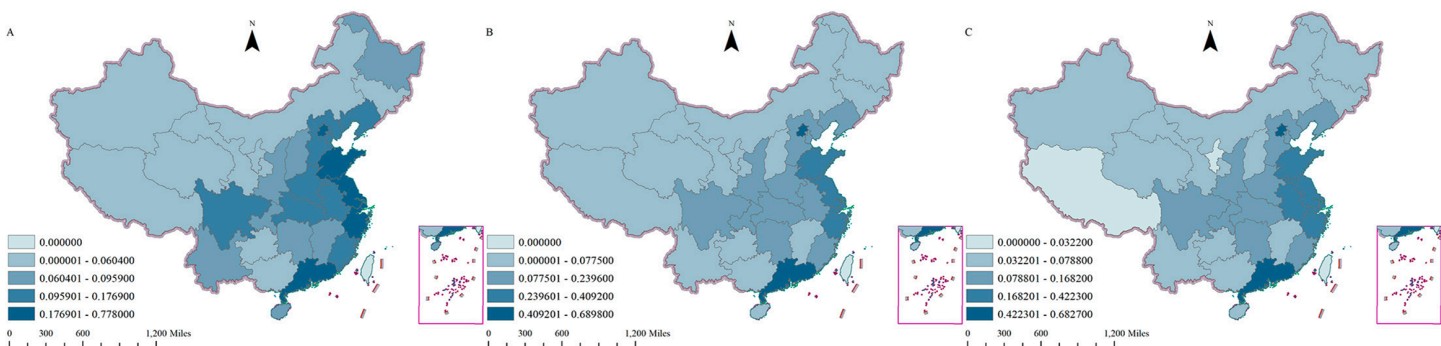

**Fig 3. Spatial distribution characteristics of overall E-commerce Development in China.** (A) Spatial distribution of overall e-commerce development in China, 2013; (B) Spatial distribution of overall e-commerce development in China, 2018; (C) Spatial distribution of overall e-commerce development in China, 2023. Maps are derived from a free and open-source website: https://cloudcenter.tianditu.gov.cn/dataSource).

in their e-commerce development. A similar trend is observed in the central and western regions, where a gradual intensification of color indicates an increase in e-commerce development, particularly in the central regions such as Henan and Hubei, where the development has been particularly robust, reaching the interval of 0.046501 - 0.077500. These findings suggest that from 2013 to 2018, e-commerce development in China experienced a substantial nationwide expansion, with the central and western regions rapidly approaching the levels of the eastern regions.

In 2023 (C), the eastern coastal region maintains its lead in e-commerce development, though the intensity of the color remains relatively unchanged, suggesting that the region has entered a period of relative stability in its development. Concurrently, the e-commerce landscape in the central and western regions has shown signs of advancement, as evidenced by an intensification of the color in these regions, particularly in western areas such as Sichuan and Chongqing, where there has been a substantial surge in e-commerce activity, reaching levels between 0.060601 and 0.078800.Overall, during the period spanning from 2013 to 2023, China's e-commerce development has exhibited notable growth, with a gradual convergence between regions. However, the eastern region continues to maintain a competitive edge.

From 2013 to 2023, China witnessed a substantial surge in its e-commerce development across the nation. The eastern coastal region has historically led the nation in e-commerce development; however, the central and western regions have also shown notable progress, narrowing the gap with the eastern region over the past decade. This indicates that China has achieved substantial progress in e-commerce development and that the imbalance between regions is progressively diminishing.

## Results of the Dagum Gini coefficient

From 2013 to 2023, the overall Gini coefficient fluctuates between 0.471 (2021) and 0.505 (2013) (as shown in Table 3 ). The overall Gini coefficient is highest in 2013, indicating that the regional differences in the overall level of e-commerce development among China's provinces were greatest in 2013. Thereafter, the overall Gini coefficient underwent a slight decrease, but rebounded to 0.484 in 2022, and further declined to 0.481 in 2023.

The overall Gini coefficient demonstrates relative stability during the 11-year period, ranging from 0.4 to 0.5, indicating a noticeable regional disparity in the extent of e-commerce

**Table 3. Results of Dagum Gini coefficient and contribution rate.**

| Year | Dagum Gini coefficient | | | | contribution rate(%) | | |
|---|---|---|---|---|---|---|---|
| | Overall Gini coefficient | $G^w$ | $G^b$ | $G^t$ | $G^w$ | $G^b$ | $G^t$ |
| 2013 | 0.505 | 0.094 | 0.399 | 0.013 | 18.532 | 78.990 | 2.477 |
| 2014 | 0.488 | 0.094 | 0.379 | 0.015 | 19.252 | 77.673 | 3.075 |
| 2015 | 0.474 | 0.094 | 0.363 | 0.018 | 19.743 | 76.443 | 3.813 |
| 2016 | 0.478 | 0.094 | 0.362 | 0.022 | 19.632 | 75.693 | 4.675 |
| 2017 | 0.479 | 0.092 | 0.364 | 0.023 | 19.258 | 75.925 | 4.816 |
| 2018 | 0.477 | 0.095 | 0.355 | 0.026 | 19.987 | 74.469 | 5.543 |
| 2019 | 0.478 | 0.100 | 0.348 | 0.029 | 20.947 | 72.905 | 6.149 |
| 2020 | 0.469 | 0.099 | 0.343 | 0.027 | 21.059 | 73.103 | 5.838 |
| 2021 | 0.471 | 0.097 | 0.352 | 0.022 | 20.594 | 74.745 | 4.661 |
| 2022 | 0.484 | 0.101 | 0.357 | 0.026 | 20.931 | 73.768 | 5.302 |
| 2023 | 0.481 | 0.103 | 0.345 | 0.033 | 21.335 | 71.766 | 6.900 |

development across China's provinces. However, this disparity does not exhibit any marked deterioration or substantial improvement.

In the context of long-term trends, the Gini coefficient between groups exhibits a gradual decline, though it experiences minor fluctuations. This suggests that the disparity in e-commerce development between different groups is gradually narrowing.

As illustrated in Table 4 , the Within-group Gini coefficient in the eastern region exhibits fluctuations between 0.334 (2016) and 0.365 (2023). While there is some variability, the coefficient remains relatively stable, suggesting that the disparities in the degree of e-commerce development among the provinces within the eastern region have not exhibited a substantial tendency to widen or narrow over the course of this 11-year period. Similarly, the Within-group Gini coefficient in the central region fluctuates between 0.131 (2013) and 0.257 (2023). It is evident that the disparities within the central region exhibit a marked increasing trend, particularly from 2013 to 2023, during which the Within-group Gini coefficient escalates from 0.131 to 0.257, signifying an escalating imbalance in the e-commerce development levels of the provinces within the central region. The fluctuations of the Within-group Gini coefficient in the western region range from 0.236 (2013) to 0.304 (2022). The western region has undergone changes in its differences, although these fluctuations are comparatively minimal in relation to those observed in the central region. Nonetheless, these fluctuations do indicate a certain degree of instability, suggesting the presence of differential changes in the level of e-commerce development within the western region. The Within-group Gini coefficient of the northeastern region fluctuates between 0.171 (2023) and 0.210 (2018). The fluctuations during this 11-year period are negligible, suggesting that the level of e-commerce development in the provinces of the Northeast region is relatively homogeneous.

The Between-group Gini coefficient between the eastern and central regions fluctuates between 0.482 (2023) and 0.581 (2013). The long-term trend indicates a gradual decline, suggesting a narrowing of the disparity in e-commerce development between the eastern and central regions. However, the overall disparity remains substantial. The Between-group Gini coefficient between the eastern and western regions exhibits variability, ranging from 0.616 (2013, 2015) to 0.721 (2013). This value is high and fluctuating, indicating that the difference in the level of e-commerce development between the eastern and western regions is more significant, and despite small fluctuations, the overall difference is significant; the intergroup Gini coefficient between the eastern and northeastern regions fluctuates between 0.632 (2019) - 0.679 (2021). The coefficient indicates a substantial disparity in the level of e-commerce

**Table 4. Results of the decomposition of the differences in Dagum Gini coefficient.**

| Year | Within-group Gini coefficient | | | | Between-group Gini coefficient | | | | | |
|---|---|---|---|---|---|---|---|---|---|---|
| | Eastern Region | Central Region | Western Region | Northeastern Region | Eastern & Central | Eastern & Western | Eastern & Northeastern | Central & Western | Central & Northeastern | Western & Northeastern |
| 2013 | 0.344 | 0.131 | 0.236 | 0.192 | 0.581 | 0.721 | 0.616 | 0.291 | 0.180 | 0.274 |
| 2014 | 0.342 | 0.146 | 0.261 | 0.179 | 0.549 | 0.693 | 0.606 | 0.301 | 0.187 | 0.265 |
| 2015 | 0.342 | 0.174 | 0.247 | 0.196 | 0.520 | 0.666 | 0.616 | 0.304 | 0.242 | 0.246 |
| 2016 | 0.334 | 0.176 | 0.265 | 0.198 | 0.533 | 0.661 | 0.655 | 0.294 | 0.268 | 0.250 |
| 2017 | 0.322 | 0.175 | 0.282 | 0.183 | 0.537 | 0.664 | 0.657 | 0.302 | 0.260 | 0.252 |
| 2018 | 0.339 | 0.162 | 0.287 | 0.210 | 0.517 | 0.659 | 0.651 | 0.315 | 0.276 | 0.263 |
| 2019 | 0.356 | 0.177 | 0.303 | 0.203 | 0.510 | 0.648 | 0.650 | 0.323 | 0.292 | 0.274 |
| 2020 | 0.352 | 0.168 | 0.297 | 0.196 | 0.504 | 0.636 | 0.645 | 0.308 | 0.282 | 0.265 |
| 2021 | 0.341 | 0.165 | 0.291 | 0.203 | 0.518 | 0.633 | 0.679 | 0.287 | 0.311 | 0.280 |
| 2022 | 0.357 | 0.164 | 0.304 | 0.197 | 0.538 | 0.648 | 0.670 | 0.293 | 0.279 | 0.274 |
| 2023 | 0.365 | 0.257 | 0.300 | 0.171 | 0.482 | 0.652 | 0.650 | 0.373 | 0.352 | 0.257 |

development between the eastern and northeastern regions, with no discernible trend of narrowing over the 11-year period under consideration. In contrast, the coefficient for the central and western regions fluctuates between 0.291 (2013) and 0.373 (2023). This coefficient demonstrates an upward trend, suggesting that the disparity in the level of e-commerce development between the central and western regions is gradually expanding. In the case of the Central and Northeastern region, the Between-group Gini coefficient fluctuates between 0.180 (2013) and 0.352 (2023). This indicates an upward trend, suggesting an expansion in the gap between the levels of e-commerce development between the central and northeastern regions. The intergroup Gini coefficient between the western and northeastern regions fluctuates between 0.257 (2023) and 0.292 (2019). While these fluctuations are minimal, discernible disparities do exist.

From 2013 to 2023, the degree of e-commerce development in China's provinces exhibited significant variations both within and between regions. A discernible trend emerged, characterized by the escalation of disparities within the central region and the maintenance of relative stability within the eastern region. Examining inter-regional disparities, the divergence between the eastern region and other regions, including the western and northeastern regions, became increasingly pronounced and underwent a gradual decline. Conversely, the disparities between the central region and the western region, as well as between the central region and the northeastern region, exhibited an upward trajectory. In summary, the disparity in the degree of e-commerce development among Chinese provinces remains pronounced on a regional scale.

## Results of the Moran's I

**Global Moran's I.** As demonstrated in Table 5, the global Moran's I of China's overall e-commerce development level from 2013 to 2023 has passed the significance test, indicating spatial autocorrelation. The results of the global Moran's I demonstrate that the e-commerce development level of each province is not randomly distributed, but exhibits clear spatial clustering characteristics. Consequently, further examination of the distribution pattern and characteristics of spatial clustering of e-commerce development level in China is warranted, utilizing the local Moran's I.

**Local Moran's I.** An analysis of China's e-commerce development reveals notable geographic disparities and evolving trends between 2013 and 2023, as illustrated in Fig 4. In general, regions exhibiting higher levels of e-commerce development are predominantly situated

**Table 5. Overall Development Level of E-Commerce Global Moran's I from 2013-2023.**

| Year | I | std(I) | z-value | p-value |
|---|---|---|---|---|
| 2013 | 0.1173** | 0.05160 | 2.8468 | 0.012 |
| 2014 | 0.1041** | 0.05410 | 2.5399 | 0.014 |
| 2015 | 0.0949** | 0.05230 | 2.4313 | 0.018 |
| 2016 | 0.0994** | 0.05393 | 2.4619 | 0.019 |
| 2017 | 0.1042** | 0.05320 | 2.5362 | 0.015 |
| 2018 | 0.0920** | 0.05340 | 2.3672 | 0.020 |
| 2019 | 0.0790** | 0.05210 | 2.1708 | 0.029 |
| 2020 | 0.0795** | 0.05250 | 2.1402 | 0.027 |
| 2021 | 0.0787** | 0.05250 | 2.1305 | 0.026 |
| 2022 | 0.0770** | 0.05040 | 2.1747 | 0.029 |
| 2023 | 0.1088** | 0.05500 | 2.5821 | 0.014 |

Note: ***, **, * indicate significant at 1%, 5%, 10% level, respectively.

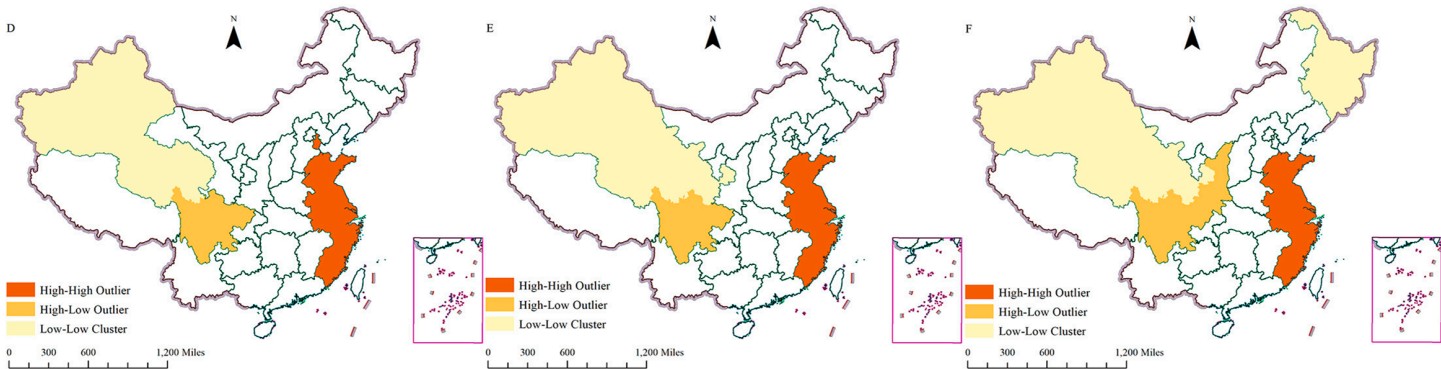

**Fig 4. The Lisa cluster map of China's overall e-commerce development level.** (D shows the LISA cluster map of China's overall e-commerce development level in 2013; E shows the LISA cluster map of China's overall e-commerce development level in 2018; and F shows the LISA cluster map of China's overall e-commerce development level in 2023, with the maps taken from a free and open source website: https://cloudcenter.tianditu.gov.cn/dataSource).

in the eastern coastal areas, exhibiting a decline in development from coastal regions towards inland areas.

As illustrated by the Lisa cluster map, in the three-year period under consideration, regions exhibiting high levels of e-commerce development are predominantly located in the eastern coastal region, including Beijing, Shanghai, and Guangdong. These regions are distinguished by their economic development, advanced infrastructure, and high Internet penetration rates, which collectively create a conducive environment for the proliferation of e-commerce. Conversely, the central and western regions exhibited comparatively lower levels of e-commerce development. A notable disparity in the levels of e-commerce development was observed between 2013 and 2018. In 2013, the eastern coastal region dominated, while in 2018, e-commerce development had penetrated the central and western regions, though the eastern coastal region maintained its dominance. By 2023, e-commerce development had further advanced into the central and western regions, with some inland regions also experiencing an increase in their development levels.

A close overall look at China's e-commerce development shows a trend of gradual diffusion from the east coast to the central and western regions. While the eastern coastal region continues to dominate, the central and western regions are rapidly gaining ground. This observed geographical differentiation and evolutionary trend in can be attributed to

a complex interplay of various interconnected factors. Firstly, policy support, encompassing targeted regional development strategies, fiscal incentives, and regulatory frameworks, has played a pivotal role in shaping uneven growth trajectories and fostering specific e-commerce hubs or lagging regions over time. Secondly, infrastructure development, particularly advancements in digital connectivity and logistics infrastructure, directly determines the physical and digital accessibility of e-commerce services, thereby creating significant spatial disparities. Furthermore, the varying rates of Internet penetration, reflecting not only access but also affordability and digital literacy, critically influence the size and engagement level of the online consumer base across different regions. Lastly, dynamic changes in consumer attitudes and purchasing habits, driven by increasing trust in online platforms, a preference for convenience, and evolving digital literacy, have collectively fueled the overall evolutionary trajectory of e-commerce adoption and its spatial diffusion, leading to a nuanced landscape of development.

## Results of the convergence analysis

$\sigma$-**convergence.** The coefficient of variation (CV) method was employed to assess the $\sigma$-convergence of e-commerce development levels across the country, as well as within the eastern, central, western, and northeastern regions ( as shown in Fig 5), it is evident that a pronounced $\sigma$-convergence trend is observed nationwide. The eastern region exhibits stability, devoid of any discernible convergence or divergence. The central region experiences an initial widening of disparities, followed by a subsequent convergence. The western region demonstrates erratic fluctuations in internal disparities. The northeastern region displays convergence during the initial period, subsequently accompanied by a widening of variance in the subsequent period. The coefficient of variation for the entire country attained its maximum value in 2013, approaching 1.2, and subsequently exhibited an annual decline, approaching 1 by 2023. This declining trend signifies a nationwide $\sigma$-convergence phenomenon, indicating a gradual narrowing of the disparities among regions. The coefficient of

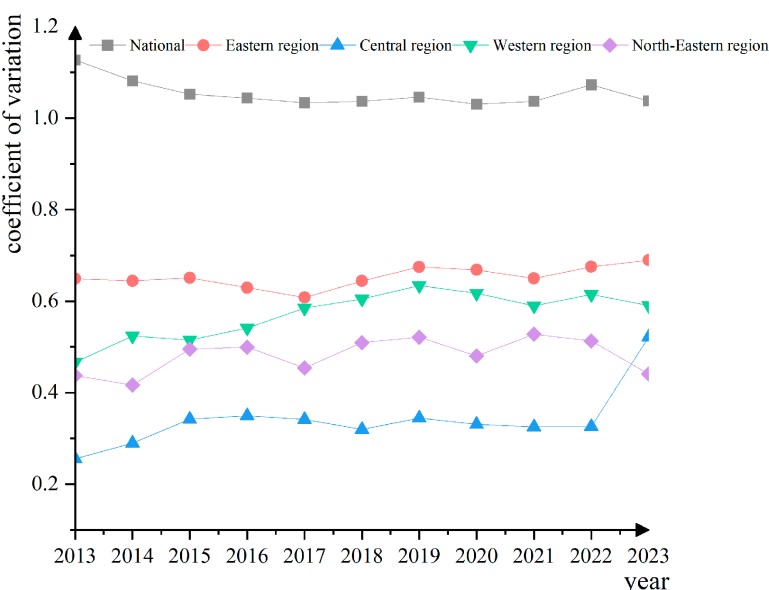

**Fig 5. Coefficient of variation of the level of e-commerce development in China.**

variation in the eastern region exhibited relative stability during the period 2013-2023, fluctuating within the range of 0.7 to 0.8. This finding suggests that the variations observed among the regions within the eastern region do not exhibit a discernible trend during the specified period. Additionally, it indicates an absence of significant $\sigma$-convergence or divergence phenomena.

The coefficient of variation for the central region demonstrates an increasing trend that subsequently transitions into a decreasing trend over the observed period from 2013 to 2023.The coefficient of variation exhibits a gradual increase from 2013 to 2017, with a rise from approximately 0.2 to nearly 0.4.Following 2017, a decline is initiated, leading to a value close to 0.3 by 2023. This trend signifies that the central region initially experienced an expansion in internal variation during the initial period, subsequently exhibiting a degree of convergence in the subsequent period. The coefficient of variation in the western region exhibited a decreasing trend, followed by an increasing trend, and then a subsequent decreasing trend during the period from 2013 to 2023. The coefficient of variation demonstrated a decline from approximately 0.4 to approximately 0.3 during the years 2013 to 2015, an increase to approximately 0.5 in the period from 2015 to 2019, and a subsequent gradual decrease after 2019. This intricate trend suggests that the variation within the western region is more unstable, with no discernible $\sigma$-convergence or divergence trend. The coefficient of variation in the Northeast region exhibited a decreasing trend, followed by an increasing trend, during the period 2013-2023. The coefficient of variation decreased from approximately 0.4 to approximately 0.3 in the 2013-2019 period, and subsequently increased to approximately 0.4 after 2019. This trend suggests that the Northeast region experienced a degree of $\sigma$-convergence in the early period, followed by a widening of internal differences in the later period. The internal variation subsequently widened.

**Absolute $\beta$-convergence analysis.** As illustrated in Table 6, the spatial error model indicates the presence of significant spatial autocorrelation, as evidenced by both Moran's I and the Lagrange multiplier test. However, the robustness test (Robust Lagrange multiplier) suggests that this correlation may not be substantial. In the spatial lag model, the Lagrange multiplier test indicates the presence of significant spatial autocorrelation; however, the robustness test (Robust Lagrange multiplier) suggests that this correlation may not be significant. Consequently, the OLS model was selected.

As illustrated in Table 7, the OLS model run for the absolute $\beta$-convergence analysis of China's e-commerce development level indicates that the coefficient of 1_score (initial e-commerce development level) is –0.0176644, the standard error is 0.0062189, the t-value is –2.84, and the P-value is 0.005. The findings indicate that for each unit increase in the initial e-commerce development level, the subsequent e-commerce development speed will decrease by 0.0176644 units. The p-value is less than 0.05, indicating that the coefficient is statistically significant. The initial level of development exerts a significant negative influence on the subsequent speed of development, indicating that there is an absolute $\beta$-convergence of the level of e-commerce development in China. Concurrently, the data also indicate that the regions

**Table 6. Results of the Lagrange test.**

| Test | | Statistic | df | P-value |
|---|---|---|---|---|
| Spatial error | Moran'I | 2.171 | 1 | 0.030 |
| | Lagrange multiplier | 4.241 | 1 | 0.039 |
| | Robust Lagrange multiplier | 0.260 | 1 | 0.610 |
| Spatial lag | Lagrange multiplier | 4.001 | 1 | 0.045 |
| | Robust Lagrange multiplier | 0.020 | 1 | 0.886 |

**Table 7. Results of the OLS.**

| d_score | Coefficient | Std.err. | t | P > |t| | [95% conf.interval] | |
|---|---|---|---|---|---|---|
| l_score | −.0176644 | .0062189 | −2.84 | 0.005 | −.0299012 | −.0054275 |
| _cons | .0023988 | .0015486 | 1.55 | 0.122 | −.0006484 | .0054459 |

that initiated later have faster e-commerce development and are rapidly catching up with the pioneers.

**Conditional $\beta$-convergence analysis.** As illustrated in Table 8, the results of the Lagrange test for the level of e-commerce development in China and the explanatory variables indicate that, in the spatial error model, Moran's I, Lagrange multiplier, and Robust Lagrange multiplier tests all demonstrate significant spatial autocorrelation. In the spatial lag model, both Lagrange multiplier and Robust Lagrange multiplier tests exhibited significant spatial autocorrelation.

Given the observed significant spatial autocorrelation in the test results, this study employs the Spatial Durbin Model (SDM). To ascertain whether the SDM would degenerate, a Wald Test for SAR was conducted, yielding a p-value of 0.0498. This result indicates that the SDM does not degenerate. Furthermore, Likelihood Ratio (LR) tests were performed to compare the SDM with the Spatial Autoregressive Model (SAR) and the Spatial Error Model (SEM), At the 10% significance level, the results demonstrate a superior goodness-of-fit for the SDM. Consequently, the SDM is preferentially selected for this analysis.

The results of the SDM model run, as presented in Table 9, reveal a direct effect characterized by a significant positive relationship between economic development and e-commerce development. The regression coefficient (ln_gdp) of 0.0710209 and a P-value of 0.010 indicate statistical significance, suggesting that economic development in a region has a substantial positive influence on its e-commerce development. This indicates that as a region's economic development level increases, the level of e-commerce development in the region also tends to rise. The ln_county coefficient is 0.0699656, and the P-value is 0.089, indicating that it is close to being significant at the 10% level. This suggests that the county's economic development has a positive impact on e-commerce development in the region. In the spatial spillover effect, the ln_finance coefficient is −0.0177471, with a P-value of 0.089, which is close to significant at the 10% level of significance, indicating that the level of financial development in neighboring regions exerts a negative impact on the level of e-commerce development in this region. The ln_county coefficient is 0.20. 90297, with a P-value of 0.006, which is significant, indicating that the neighboring regions' level of county economic development has a significant positive impact on the level of e-commerce development in this region, i.e., the more developed the county economy in the neighboring regions, the higher the level of e-commerce development in this region, and there is a significant spatial spillover effect.

The preceding analysis demonstrates a significant positive correlation between regional economic development and e-commerce development, indicating economic development as a

**Table 8. Results of the Lagrange test.**

| Test | | Statistic | df | P-value |
|---|---|---|---|---|
| Spatial error | Lagrange multiplier | 43.444 | 1 | 0.000 |
| | Robust Lagrange multiplier | 50.894 | 1 | 0.000 |
| Spatial lag | Lagrange multiplier | 6.650 | 1 | 0.010 |
| | Robust Lagrange multiplier | 14.100 | 1 | 0.000 |

**Table 9. Results of the SDM model.**

| | Score | Coefficient | Std.err. | z | P > \|t\| | [95% conf.interval] | |
|---|---|---|---|---|---|---|---|
| Main | ln_finance | .0001391 | .0056668 | 0.02 | 0.980 | −.0109676 | .0112459 |
| | ln_gdp | .0710209 | .0275432 | 2.58 | 0.010 | .0170371 | .1250046 |
| | ln_county | .0699656 | .041077 | 1.70 | 0.089 | −.0105438 | .150475 |
| | ln_edu | −.0159715 | .0168046 | −0.95 | 0.342 | −.0489079 | .016965 |
| Wx | ln_finance | −.0177471 | .010449 | −1.70 | 0.089 | −.0382268 | .0027326 |
| | ln_gdp | −.0719553 | .0578457 | −1.24 | 0.214 | −.1853308 | .0414203 |
| | ln_county | .2090297 | .0766836 | 2.73 | 0.006 | .0587326 | .3593268 |
| | ln_edu | −.0363956 | .0351092 | −1.04 | 0.300 | −.1052083 | .0324172 |
| Spatial | rho | −.0711557 | .0802863 | −0.89 | 0.375 | −.228514 | .0862026 |
| Variance | sigma2_e | .0003552 | .0000272 | 13.05 | 0.000 | .0003018 | .0004085 |

crucial driver. Notably, a spatial spillover effect of county-level economic development exists, whereby neighboring regions' economic prosperity exerts a significant positive influence on local e-commerce development. This suggests that interregional economic integration, facilitated by e-commerce, fosters interconnected growth through mechanisms such as logistics integration, supply chain optimization, and consumer market expansion.

This phenomenon can be attributed to: (1) the direct impact of local economic development on e-commerce adoption, driven by increased income, enhanced consumer demand, and robust infrastructure; (2) the market integration effect, where e-commerce dismantles geographical barriers, promoting cross-regional economic interactions. Specifically, regional economic growth stimulates contiguous consumption and production, while improved logistics and supply chains, developed in one region, positively impact neighboring areas through network interconnections. Furthermore, successful e-commerce strategies in adjacent counties serve as models for emulation, while competitive pressures incentivize accelerated local e-commerce development.

## Discussion

### Interpretation of results

This study utilizes systematic data analysis and scientific model validation to thoroughly explore the spatial and temporal dynamic characteristics of e-commerce development in China and its convergence trend. The results of the study reveal the heterogeneity of e-commerce in different regions and analyze the driving factors and spatial correlation effects behind it from a multi-dimensional perspective. The results of the study demonstrate that:

The extant research data demonstrate that the level of e-commerce development in China is characterized by significant imbalance among regions, with this difference exhibiting a certain degree of continuity over time. However, the data also reveal a gradual trend of dynamic adjustment in the spatial distribution of e-commerce development. Specifically, the eastern coastal region has historically occupied the leading position in e-commerce development over the past decade, a position attributable to its unique geographical advantages, well-developed digital infrastructure, intensive economic activities, and early policy support. For instance, provinces such as Zhejiang, Guangdong, and Jiangsu have demonstrated leadership in e-commerce transaction scale, underpinned by robust industrial clusters and logistics networks. These regions have also exhibited notable competitive advantages in the domains of platform economy and digital innovation. Conversely, the central and western regions have historically lagged behind in terms of e-commerce development, particularly in its initial stages.

These regions exhibit a deficiency in fundamental infrastructure, as evidenced by the underdeveloped Internet infrastructure, inefficient logistics and distribution systems, and the limited engagement of market participants.

Recent years have witnessed a noteworthy catching-up trend in the development of e-commerce in the central and western regions. This phenomenon can be attributed to a multitude of factors. Primarily, policy guidance at the national level has been instrumental in this regard. For instance, the implementation of the "Internet Plus" strategy and the rural revitalization policy has expedited the development of digital infrastructure in these regions. Secondly, the strategic decisions of e-commerce platform enterprises, such as the adoption of a downward strategy, have further augmented the market potential of the central and western regions. This has led to the successful attraction of a substantial number of small and medium-sized merchants and consumers to participate in the e-commerce ecosystem. Moreover, the presence of lower labor costs and abundant land resources in the central and western regions has fostered the establishment of e-commerce-related industries. Consequently, these factors have contributed to the rapid increase in the scale and activity of e-commerce transactions in the central and western regions. Nevertheless, the inter-regional gap still exists and is difficult to be completely eliminated in the short term. The eastern region still dominates the high-end e-commerce field by virtue of its first-mover advantage and technological accumulation, while the central and western regions rely more on policy dividends and basic market expansion. To address these imbalances and foster more equitable e-commerce growth in the future, sustained efforts must be made to enhance infrastructure, facilitate technology diffusion, and cultivate talent.

This study utilizes an econometric model to verify the nationwide convergence trend of China's e-commerce development level and conducts an in-depth analysis of the $\sigma$-convergence and $\beta$-convergence dimensions. The results of the $\sigma$-convergence analysis indicate that the dispersion of e-commerce development level among different regions gradually decreases, i.e., the gap shows a narrowing trend on the whole, while $\beta$-convergence analysis results show that the growth rate of the later regions is significantly higher than that of the leading regions, thus gradually approaching the leading level. Specifically, the western and central provinces are particularly outstanding in terms of e-commerce development speed. For instance, Guizhou, Sichuan, and Shaanxi provinces have exhibited substantially higher e-commerce turnover growth compared to the eastern region in recent years. The emergence of this latecomer advantage is intricately associated with policy favoritism and resource investment. Guizhou Province's adoption of the "big data + e-commerce" model has led to notable advancements in the digitization capabilities of local enterprises, attracting external capital and technology. However, the convergence process has been marked by challenges, resulting in short-term volatility.

This study also finds that the development of e-commerce in China has a significant spatial spillover effect, i.e., the level of e-commerce development in a certain region positively affects its neighboring regions through economic ties and information flows. The existence of this effect highlights the importance of regional cooperation and integrated market construction in enhancing the overall e-commerce development level. The analytical results of the spatial econometric model indicate that e-commerce development in the eastern coastal region exerts a significant radiation effect on neighboring provinces through the supply chain network, logistics channels, and the spillover of consumer demand. In contrast, the central and western regions exhibit a spatial spillover effect, albeit with comparatively limited intensity and scope. This phenomenon is attributed to the decentralized nature of infrastructure and the absence of robust economic interconnections in these regions. However, with the augmentation of high-speed rail networks and the enhancement of logistics efficiency, there has

been a gradual escalation in the spatial connectivity of the central and western regions. The augmentation of this spatial spillover effect offers novel prospects for interregional synergistic development.

## Limitations of the study

Despite the comprehensive spatial distribution and convergence analysis provided by this study, there are still some limitations that require refinement in future studies:

First, the present study elects to utilize a select number of pivotal variables (e.g., e-commerce transaction volume, Internet penetration rate) as proxies for the assessment of developmental level. However, it does not adequately address the influence of soft factors, such as cultural influences and disparities in consumer behavior, on the evolution of e-commerce.

Second, the study employed methodologies such as the $\beta$-convergence model and the spatial Durbin model to analyze convergence trends and their influencing factors. These models presuppose the presence of a relatively homogeneous intrinsic mechanism across regions. However, they may have overlooked the intricate characteristics of certain regions, such as remote and underdeveloped areas. Concurrently, the adjacency matrix is employed to delineate regional spatial relationships, yet it falls short in fully considering the further influence of economic linkage networks and industrial chain linkage effects on regional relationships.

Finally, the present study primarily focuses on the data analysis from 2013 to 2023. While this analysis reveals the spatio-temporal dynamic characteristics of a decade, it is difficult to observe the longer-term e-commerce development pattern and the persistent influence of uncertainty factors due to the limitation of the data sampling time period.

## Conclusion and suggestions

This study utilizes a systematic approach to assess China's e-commerce development from 2013 to 2023. Employing spatial analysis and econometric modeling, it systematically examines the spatio-temporal distribution characteristics and regional development disparities. The study's primary conclusions are as follows:

First. At the national level, there has been a substantial increase in e-commerce development, with a gradual narrowing of the gap between the eastern, central, and western regions. The eastern region continues to demonstrate a leading position, while the potential of the central and western regions remains untapped.

Second. A clear trend of $\sigma$ convergence is evident on a nationwide scale. The eastern region exhibits stability, devoid of any discernible convergence or divergence. The central region experiences an expansion in its difference during the initial stage, subsequently converging in the subsequent stage. The internal difference in the western region is characterized by instability, while the northeastern region demonstrates convergence in the early stage, followed by an expansion in the late stage.

Third. The phenomenon of $\beta$-convergence indicates that regions with a late start are rapidly catching up with the pioneer regions, and that economic level, digital infrastructure construction, and policy support are the primary driving forces behind this convergence.

In order to achieve a more balanced regional development of e-commerce, this study proposes the following policy recommendations: first, to continuously increase investment in digital infrastructure construction in the central and western regions, with the aim of narrowing the hardware gap with the eastern regions; second, to optimize the policy design, with a focus on support for the cultivation of e-commerce in counties and the rural market; and third, to promote synergistic interregional development to enhance the overall efficiency

through resource integration and market linkage. The implementation of these measures is expected to not only bridge the existing regional disparities but also catalyze the sustainable growth of e-commerce in China.

## Supporting information

**S1 Data.**
(XLS)

## Author contributions

**Conceptualization:** Yuhuan Wu.

**Data curation:** Haisong Wang.

**Formal analysis:** Daqun Guo.

**Investigation:** Yan Li.

**Methodology:** Hang Zhang.

**Writing – original draft:** Tianyuan Shan.

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
