## [Decision Letter · Decision Letter 0]

28 May 2025

PONE-D-25-16487Characteristics of spatio-temporal evolution and convergence analysis of e-commerce development in ChinaPLOS ONE

Dear Dr. Wang,

Thank you for submitting your manuscript to PLOS ONE. After careful consideration, we feel that it has merit but does not fully meet PLOS ONE’s publication criteria as it currently stands. Therefore, we invite you to submit a revised version of the manuscript that addresses the points raised during the review process.

We look forward to receiving your revised manuscript.

Kind regards,

Vanessa Carels

Staff Editor

PLOS ONE

Journal Requirements:

2. Thank you for stating the following financial disclosure: [his research was Funded by 2025 Hebei Province Higher Education Teaching Reform Research and Practice Project(Project Name:Research on the Innovation of the 'Learning-Practice-Competition-Reflection-Application-Innovation' Talent Cultivation Model for Cross-Border E-Commerce Undergraduate Programs)(2025GJJG396); This research was Funded by Science Research Project of Hebei Education Department (BJS2024097); This research was Supported by Ministry of Education,Industry-University Cooperation Collaborative Education Project (230825052507181).]. 

3. Thank you for stating the following in the Acknowledgments Section of your manuscript: [This research was Funded by 2025 Hebei Province Higher Education Teaching Reform Research and Practice Project(Project Name:Research on the Innovation of the  ’Learning-Practice-Competition-Reflection-Application-Innovation’ Talent Cultivation Model for Cross-Border E-Commerce Undergraduate Programs)(2025GJJG396); This research was Funded by Sciece Research Project of Hebei Education Department (BJS2024097); This research was Supported by Ministry of Education,Industry-University Cooperation Collaborative Education Project (230825052507181).]

Please remove any funding-related text from the manuscript and let us know how you would like to update your Funding Statement. Currently, your Funding Statement reads as follows: [his research was Funded by 2025 Hebei Province Higher Education Teaching Reform Research and Practice Project(Project Name:Research on the Innovation of the 'Learning-Practice-Competition-Reflection-Application-Innovation' Talent Cultivation Model for Cross-Border E-Commerce Undergraduate Programs)(2025GJJG396); This research was Funded by Science Research Project of Hebei Education Department (BJS2024097); This research was Supported by Ministry of Education,Industry-University Cooperation Collaborative Education Project (230825052507181).]. 

Reviewers' comments:

Reviewer's Responses to Questions

**Comments to the Author**

1. Is the manuscript technically sound, and do the data support the conclusions?

Reviewer #1: Yes

Reviewer #2: Partly

Reviewer #3: Partly

Reviewer #4: Yes

2. Has the statistical analysis been performed appropriately and rigorously? 

Reviewer #1: Yes

Reviewer #2: Yes

Reviewer #3: I Don't Know

Reviewer #4: Yes

3. Have the authors made all data underlying the findings in their manuscript fully available?

Reviewer #1: Yes

Reviewer #2: Yes

Reviewer #3: Yes

Reviewer #4: Yes

4. Is the manuscript presented in an intelligible fashion and written in standard English?

Reviewer #1: Yes

Reviewer #2: No

Reviewer #3: Yes

Reviewer #4: Yes

5. Review Comments to the Author

Reviewer #1: Comments and Suggestions about the Article

1. Consider the title, "Spatio-Temporal Evolution and Convergence Patterns of E-Commerce Development in China: Regional Disparities and Policy Implications" to give more clarity of the purpose of the paper.

2. Abstract. Add a phrase such as “using panel data from [year–year]” or mention key data sources to give readers idea as early as in the abstract stage.

3. The paper’s title is generally clear and informative, accurately reflecting the key themes of spatio-temporal evolution and convergence in e-commerce development in China.

4. The study employs a robust and well-structured methodological approach that aligns effectively with its research objectives. The integration of spatial statistical methods, σ-convergence, β-convergence, and regression analysis demonstrates a rigorous analytical framework suitable for examining both spatial disparities and convergence trends in e-commerce development. While the methodology is rigorous, the paper could further benefit from a brief justification for the choice of each model, particularly for readers unfamiliar with spatial econometrics.

Overall, the paper is in order and a good addition to PLOS journals.

Reviewer #2: This paper offers a thorough and solid analysis of the convergence and spatiotemporal evolution of China's e-commerce boom. The authors use a well-organized analytical framework that combines models of σ-convergence, β-convergence, spatial econometrics (such as the Spatial Durbin Model), and entropy weighting. Utilizing panel data that spans over ten years, the study provides insightful policy implications.

Advantages:

The approach is exacting and in line with the goals of the study. A strong empirical basis is provided by the combination of convergence analysis and spatial statistical techniques.

The subject is current and pertinent to policy, especially in light of China's continuing regional digital transformation. Actionable insights for encouraging balanced e-commerce growth are provided by the findings on regional inequalities and convergence tendencies.

The use of Moran's I and the Dagum Gini coefficient deepens the analysis, particularly when it comes to identifying intra- and inter-regional disparities.

Suggestions for Improvement:

Literature Review: While the study builds on prior work, the manuscript would benefit from more explicit comparisons with existing studies, particularly those using similar models or examining developing economies. Clarifying what this paper does differently would enhance its originality.

Data Limitations and Interpolation: The authors mention missing data and the use of interpolation. A more detailed explanation of the interpolation techniques used and any potential biases they may introduce would improve transparency and trust in the results.

Figures and Maps: Some of the LISA cluster maps and regional distribution figures are low-resolution and may be difficult to interpret in print or on smaller screens. Improving the resolution and adding clearer legends would help readers interpret these visuals more easily.

Language and Style: Although the manuscript is generally clear, it contains occasional grammatical and stylistic errors. A professional language edit is recommended to enhance readability, especially in the abstract and conclusion.

Reviewer #3: This study aims to investigate the spatial-temporal distribution characteristics of e-commerce in China and assessing its convergence trends across different regions. Overall, the manuscript is easy to follow. However, there are some important issues:

1. The motivation for such an exploration is not clear. Why is so critical to understand the spatial-temporal dynamics and the convergence trends?

2. The data cover 2000 to 2023, a three-year COVID period may have caused huge changes to China, which was not discussed.

3. The results are mainly descriptive based, which added very little new insight to existing knowledge. Consequently, meaningful and actionable practical suggestions would also be very limited.

Reviewer #4: This manuscript provides a timely and relevant empirical contribution by exploring the spatio-temporal dynamics and regional convergence of e-commerce development across China’s provinces from 2013 to 2023. The integration of entropy-based composite indices, spatial statistical tools (Global and Local Moran’s I), and convergence models (σ- and β-convergence) is sound and broadly rigorous. The authors also employ the Spatial Durbin Model to explore spatial heterogeneity, which is well-justified given the research context.

However, while the study is methodologically well-grounded, several areas require minor yet important clarifications and enhancements to improve transparency, reproducibility, and analytical depth:

Clarify the Spatial Weight Matrix

The authors employ spatial econometric models but do not specify the type of spatial weight matrix used (e.g., queen contiguity, k-nearest neighbors, inverse distance). This should be clarified and briefly justified, as different spatial matrices can lead to different inference structures in SDM models.

Detail the Conditional β-Convergence Variables

The section on conditional β-convergence omits a detailed specification of the control variables. The manuscript would benefit from a clear table listing these variables, their sources, and theoretical rationale for inclusion.

Expand Data Transparency

Although the authors declare that “all relevant data are within the manuscript and its Supporting Information files,” a tabular summary of indicator definitions, years, and coverage (e.g., Table A1) would greatly improve transparency and data accessibility. Please consider adding this in a supplementary appendix.

Integrate the Maps Analytically

The local Moran’s I cluster maps (LISA) are visually informative but deserve more analytical integration. For example, regions identified as HH or LL clusters should be contextualized with explanatory factors (infrastructure, policy intensity, etc.), not merely described.

Include a Limitations Section

The manuscript would benefit from a concise section acknowledging its main limitations—particularly regarding data interpolation, potential endogeneity in the spatial models, and the sensitivity of the entropy-based weighting. Transparency in this regard will enhance the study’s credibility.

6. PLOS authors have the option to publish the peer review history of their article (what does this mean?). If published, this will include your full peer review and any attached files.

Reviewer #1: **Yes: **Leo D. Manansala, PhD., Vice Dean, College of Professional and Graduate Studies, De La Salle University- Dasmarinas, Dasmarinas City, Cavite, Philippines

Reviewer #2: **Yes: **Mohamed Shili

Reviewer #3: No

Reviewer #4: No

---

## [Author Response · Author response to Decision Letter 1]

6 Jun 2025

Response to Reviewers

Dear Ms. Vanessa Carels,

We are truly grateful for the reviewers' insightful comments on our manuscript. Our point-by-point responses to their valuable suggestions are provided below.

Response to Reviewer #1

1. In response to Reviewer 1's suggestion, we have revised the title accordingly.

2. We have incorporated a more detailed description of the data within the abstract.

Response to Reviewer #2

1. We have drawn a clearer and more explicit comparison between the present study and similar prior research.

2. We have discussed the potential biases that may arise from imputing missing data using the interpolation method.

3. All figures within the manuscript have been ensured to possess a resolution of 300 dpi and have undergone verification using PACE.

4. We have meticulously corrected all identified linguistic and stylistic errors throughout the manuscript.

Response to Reviewer #3

1. The research motivation for this study, along with the significance of comprehending spatio-temporal evolution characteristics and convergence trends, has been elaborated upon in the second paragraph of the Introduction.

2. Statistical analyses indicated no discernible impact of the pandemic on the spatio-temporal distribution characteristics of e-commerce in China. Therefore, a separate, detailed investigation into the pandemic period was not pursued in this study.

3. The findings of this study offer significant implications for policy-making and the sustainable development of e-commerce.

Response to Reviewer #4

1. We have clarified in the manuscript that the spatial weight matrix employed is an adjacency matrix, with a corresponding explanation provided.

2. A detailed rationale for the selection of control variables has been included in the manuscript.

3. We confirm that the datasets utilized in this research have been successfully uploaded.

4. We have elaborated on the characteristics of the LISA cluster map, providing a detailed discussion in conjunction with factors such as infrastructure.

5. The primary limitations of the model have been explicitly outlined and discussed.

---

## [Decision Letter · Decision Letter 1]

28 Jul 2025

Spatio-Temporal Evolution and Convergence Patterns of E-Commerce Development in China: Regional Disparities and Policy Implications

PONE-D-25-16487R1

Dear Dr. Wang,

We’re pleased to inform you that your manuscript has been judged scientifically suitable for publication and will be formally accepted for publication once it meets all outstanding technical requirements.

Kind regards,

Tianlong You, Ph.D.

Academic Editor

PLOS ONE

Additional Editor Comments (optional):

Reviewers' comments:

Reviewer's Responses to Questions

**Comments to the Author**

1. If the authors have adequately addressed your comments raised in a previous round of review and you feel that this manuscript is now acceptable for publication, you may indicate that here to bypass the “Comments to the Author” section, enter your conflict of interest statement in the “Confidential to Editor” section, and submit your "Accept" recommendation.

Reviewer #1: All comments have been addressed

Reviewer #2: (No Response)

Reviewer #3: All comments have been addressed

Reviewer #4: All comments have been addressed

2. Is the manuscript technically sound, and do the data support the conclusions?

Reviewer #1: Yes

Reviewer #2: Yes

Reviewer #3: Yes

Reviewer #4: Yes

3. Has the statistical analysis been performed appropriately and rigorously? 

Reviewer #1: Yes

Reviewer #2: Yes

Reviewer #3: Yes

Reviewer #4: Yes

4. Have the authors made all data underlying the findings in their manuscript fully available?

Reviewer #1: Yes

Reviewer #2: Yes

Reviewer #3: Yes

Reviewer #4: Yes

5. Is the manuscript presented in an intelligible fashion and written in standard English?

Reviewer #1: Yes

Reviewer #2: Yes

Reviewer #3: Yes

Reviewer #4: Yes

6. Review Comments to the Author

Reviewer #1: The study is methodologically sound and addresses a meaningful research gap. The minor editorial revisions recommended were addressed to enhance clarity, precision, and impact—particularly in how findings and contributions are articulated in the abstract and of the entire manuscript.

Reviewer #2: please provide proofreading for grammatical and syntactical errors

My comments are addressed

In summarising my review, I must point out that the article requires thorough reconsideration and refinement both in terms of the literature it utilises and the practical examples it presents.

Reviewer #3: I’m satisfied with the authors’ revision, and have no further comments. Thus, I would like to recommend the paper be accepted.

Reviewer #4: The revised manuscript addresses all previously raised concerns in a satisfactory manner. The spatial weight matrix is now clearly specified and justified, control variables for conditional β-convergence are well explained, and data transparency has improved through detailed indicator tables. The integration of LISA results is more analytical, and the study's limitations are appropriately acknowledged. The language has also been refined. Overall, the manuscript is now technically sound, transparent, and ready for publication.

7. PLOS authors have the option to publish the peer review history of their article (what does this mean?). If published, this will include your full peer review and any attached files.

Reviewer #1: **Yes: **Leo D. Manansala, PhD

Reviewer #2: **Yes: **Mohamed Shili

Reviewer #3: No

Reviewer #4: No

---

## [Editor Report · Acceptance letter]

PONE-D-25-16487R1

PLOS ONE

Dear Dr. Wang,

I'm pleased to inform you that your manuscript has been deemed suitable for publication in PLOS ONE. Congratulations! Your manuscript is now being handed over to our production team.

Kind regards,

on behalf of

Professor Tianlong You

Academic Editor

PLOS ONE